# Connection between Mesenchymal Stem Cells Therapy and Osteoclasts in Osteoarthritis

**DOI:** 10.3390/ijms23094693

**Published:** 2022-04-23

**Authors:** Lidia Ibáñez, Paloma Guillem-Llobat, Marta Marín, María Isabel Guillén

**Affiliations:** 1Department of Pharmacy, Cardenal Herrera-CEU Universities, 46115 Valencia, Spain; lidia.ibanez@uchceu.es (L.I.); marta.marin1@uchceu.es (M.M.); 2Department of Biomedical Science, Cardenal Herrera-CEU Universities, 46115 Valencia, Spain; paloma.guillemllobat@uchceu.es; 3Interuniversity Research Institute for Molecular Recognition and Technological Development (IDM), Polytechnic University of Valencia, Av. Vicent A. Estellés s/n, Burjasot, 46100 Valencia, Spain

**Keywords:** mesenchymal stem cells, therapy, osteoclasts, osteoarthritis

## Abstract

The use of mesenchymal stem cells constitutes a promising therapeutic approach, as it has shown beneficial effects in different pathologies. Numerous in vitro, pre-clinical, and, to a lesser extent, clinical trials have been published for osteoarthritis. Osteoarthritis is a type of arthritis that affects diarthritic joints in which the most common and studied effect is cartilage degradation. Nowadays, it is known that osteoarthritis is a disease with a very powerful inflammatory component that affects the subchondral bone and the rest of the tissues that make up the joint. This inflammatory component may induce the differentiation of osteoclasts, the bone-resorbing cells. Subchondral bone degradation has been suggested as a key process in the pathogenesis of osteoarthritis. However, very few published studies directly focus on the activity of mesenchymal stem cells on osteoclasts, contrary to what happens with other cell types of the joint, such as chondrocytes, synoviocytes, and osteoblasts. In this review, we try to gather the published bibliography in relation to the effects of mesenchymal stem cells on osteoclastogenesis. Although we find promising results, we point out the need for further studies that can support mesenchymal stem cells as a therapeutic tool for osteoclasts and their consequences on the osteoarthritic joint.

## 1. Crosstalk between Articular Cells in Osteoarthritis

### 1.1. Fundamentals of Osteoarthritis

Osteoarthritis (OA) is a chronic, inflammatory, and degenerative joint disease that affects a large part of the population, mainly older people and women, some ethnic groups, and people with lower socioeconomic status [1,2,3,4]. Although the aetiology of OA is multifactorial and not completely known, the existence of a genetic predisposition has been studied [5]. The genetic variants found contribute to susceptibility to OA, and they may become potential diagnostic and prognostic biomarkers. There are other factors that can facilitate the development of this pathology, such as intra-articular fractures that lead to post-traumatic osteoarthritis [6], obesity, and metabolic factors [7,8,9]. From the outset, obesity was considered an important risk factor for the development of knee OA, mainly for biomechanical reasons. However, it is now known that obesity also increases the risk of OA by altering metabolism and inflammation. Gender differences also exist. Studies addressing gender differences that influence the development of OA have been performed focusing on the hormonal level also on the anatomical and biomechanical characteristics of the patients. Although it is complex, it is now beginning to be understood how sexual hormones and reproductive factors influence the pathogenesis of OA, with special incidence in the knee, in addition to treatments with exogenous hormones [1,10].

Joints are made up of different tissues, such as cartilage, subchondral bone, the synovial membrane, and the infrapatellar fatty tissue, and all of them, to a greater or lesser extent, participate in the pathogenesis of the OA. The evolution of OA depends on whether it has originated in the cartilage or in the subchondral area. In OA produced in elderly patients, the problem first manifests itself in the cartilage due to the metabolic stress that the chondrocytes are suffering. However, when this pathology is caused by a fracture, there is an alteration in bone remodelling in the subchondral area prior to cartilage degradation [11,12,13,14]. Thus, articular cells and the joint microenvironment are involved in the changes that affect these two tissues and, consequently, OA pathogenesis.

### 1.2. Cellular Crosstalk in Osteoarthritis

Synovial tissue is a connective tissue that lines the joint cavity, and it produces the synovial fluid (SF) that acts as a lubricant for the joint. Two types of cell phenotypes coexist in the synovial membrane: macrophage-like synoviocytes with macrophage characteristics, and fibroblast-like synoviocytes with a structural function and responsible for producing the SF. Fibroblast-like cells are very sensitive to the changes that occur in their environment, affecting other cells in the joint, such as the osteoclasts (OCLs) and osteoblasts. Additionally, macrophages can produce osteogenic growth factors [15,16,17,18].

Articular cartilage is a connective tissue located on the surface of the diarthritic joint. The main function of this structure is to distribute and cushion the mechanical loads that occur between the most superficial layers of the cartilages of a joint, lubricating and facilitating the sliding. Moreover, hyaline cartilage is an avascular tissue where cells are in hypoxia, with a low rate of cellular proliferation and tissue regeneration. In cartilage, chondrocytes are the unique cells detected but they represent a small volume of the total tissue. These cells are responsible for renovation of extracellular matrix proteins, type II collagen, and aggrecan (hyaluronic acid, proteoglycan, and glycosaminoglycan). Chondrocytes are distributed in different layers with a different orientation and metabolism. The lower layer of the cartilage is calcified and lies just above the subchondral bone [19,20,21]. The subchondral bone separates the cartilage from the cancellous bone. This structure provides a direct interaction between chondrocytes and the cells that regulate bone remodelling, connecting the metabolism of the chondrocytes, osteoblasts, and OCLs [22,23]. It has been described that chondrocytes under mechanical stimulation could express higher levels of pro-osteoclastic factors and induced condylar subchondral bone resorption by promoting osteoclastogenesis [24].

Osteoblasts differentiate from mesenchymal stem cells (MSCs) and undergo four stages of maturation: preosteoblasts, osteoblasts, bone lining cells, and bone cells [25,26]. Osteoblast differentiation is regulated by runt-related transcription factor 2 (Runx2) and other transcription factors in the perichondrium of the endochondral bone. Runx2 acts sequentially by first inducing MSC differentiation into preosteoblasts. In a second step, it induces the expression of Sp7, which, together with Runx2 itself and the canonical Wnt signalling, induces the differentiation of preosteoblasts into immature osteoblasts. Finally, Runx2 and Sp7 act on the final maturation of osteoblasts. In addition, the proliferation of preosteoblasts is regulated by Runx2 as it induces fibroblast growth factor receptor (Fgfr) 2 and Fgfr3 [25]. Runx2 is a key transcription factor that controls both osteoblast and chondrocyte differentiation. It has been shown that there is an overexpression of this transcription factor related to the β-catenin protein in different murine models of OA, which may indicate a role in the pathogenesis of the disease [27]. It also has been shown that the phenotype and activity of OA osteoblasts in subchondral bone is altered. For example, levels of osteocalcin, receptor activator of nuclear factor kB ligand (RANKL), transforming growth factor-β1 (TGF-β1), vascular endothelial growth factor, and alkaline phosphatase activity are elevated in OA, subsequently leading to osteoclastogenesis [28,29].

OCLs are giant multinucleated cells formed by the fusion of myeloid cells, such as monocytes, macrophages, and dendritic cells. OCL generation is a multistep process mainly controlled by two factors expressed by osteoblast/stromal cells: the macrophage colony-stimulating factor (M-CSF) and RANKL [30]. M-CSF has a critical effect on the survival and proliferation of OCL precursors (OCPs) [31,32], and binding of M-CSF to its receptor (c-Fms) upregulates RANK expression on the OCP surface [33]. It is well described that the RANK/RANKL interaction induces the activation of diverse transcription factors such as NF-κB, c-Fos, and NFATc1, which regulate OCL fusion and maturation [34]. Moreover, the expression of the osteoclast-specific gene markers tartrate-resistant acid phosphatase (TRAP), cathepsin K, calcitonin receptor, and the αv-β3 integrins is also upregulated by the activation of the RANK/RANKL pathway [35]. OCLs are the only cells presenting bone-resorbing ability, which promotes the release of different osteogenic factors [28,36]. Thus, the crosstalk between OCLs and other articular cells is crucial for the maintenance of healthy joints and, since subchondral bone turnover is altered in OA, the role of these cells in the pathogenesis of OA must be addressed.

## 2. Role of Osteoclasts in Osteoarthritis

OCLs are classically known by their bone-resorbing ability. Bone removal starts when mature OCLs polarize and adhere tightly to the bone surface, creating the sealing zone [37]. Formation of the resorption lacunae begins with the release of protons and enzymes through the ruffled border membrane. Acidification of the resorption lacunae demineralizes the bone matrix [38] and secreted lysosomal proteolytic enzymes degrade organic materials [38,39]. Then, the resulting products are endocytosed and released through the functional secretory domain at the opposite side of the lacunae [40,41]. OCLs can act on other cells via paracrine factors, via bone degradation products, or via cell–cell contact. Many articles have demonstrated that several OCL-derived factors enhance MSC recruitment and osteoblast differentiation at the bone degradation area, promoting bone formation. Tang et al. [42] showed that the release of TFGβ-1 from the bone matrix induces the migration of MSCs to the site of the bone resorption. Moreover, TFGβ-1 acts on OCLs and induces Wnt and Chemokine (C-X-C motif) ligand 16 (CXCL16) release, which promote osteoblastogenesis [28,36].

The subchondral bone sits below the articular calcified cartilage and, depending on its location and properties, two forms can be distinguished: the subchondral bone plate, which is in direct contact with the cartilage, and the trabecular bone situated under the plate [43]. Various studies analysing OA patients have indicated that subchondral bone turnover in these compartments is changing depending on the phase of the pathology. Using the tibial plateau of late OA patients, Finnila et al. [44] identified differences in bone turnover between the trabecular bone and the subchondral plate, which were more evident as the OARSI grade increases. Their data showed that trabecular bone presented lower trabecular separation and higher thickness, while the subchondral plate was also thicker because of the presence of calcified cartilage [44]. In agreement with these results, reduced trabecular rods and thicker trabecular plates were described in another set of tibial plateau samples in which bone alterations were observed in both samples with damaged cartilage and samples with intact cartilage [45]. Thus, subchondral bone loss in early OA may alter the mechanical response of the joint to loading and it could drive to the cartilage damage observed in advanced disease. Moreover, bone formation may prevail in late OA, resulting in sclerosis.

Since bone resorption is only carried out by OCLs, and their activity increases osteoblast differentiation [28,36,42], recent studies have analysed the disfunction of OCLs in OA. The medial tibial plateau of OA patients presented enhanced multinucleated TRAP^+^ cells in subchondral bone [46]. However, when differences between non-sclerotic and sclerotic bone were analysed, the number of OCLs were markedly increased in sclerotic samples, and it may be related to the abundant number of inflammatory CD68^+^ macrophages [47]. These data would support that osteoclast-induced bone loss in OA may precede bone formation and sclerosis.

Besides their activity as bone-resorbing cells, OCLs have also been pointed out as mediators involved in OA pain. Symptomatic OA patients, identified by the NHANES I criterion of pain, presented higher serum TRAc5bP levels and higher TRAP^+^ OCLs density in the subchondral bone than asymptomatic OA [48]. Similar results were obtained in patients with OA pain that underwent total knee replacement, observing an increase in the OCLs density in the subchondral bone. Interestingly, these cells were found to express nerve growth factor (NGF) [49], a nociceptive pain mediator in inflammation [50].

OCLs also play a role in modulating immune responses by acting as antigen-presenting cells (APC) and activating T cells. The first evidence that OCLs act as APCs came in the early 21st Century. Rivollier et al. [51] showed that human OCLs express major histocompatibility complex (MHC) class II and the co-stimulatory molecule CD86. Later studies not only confirm the presence of this molecules on the surface of human and murine OCLs, but also the expression of chemokines and cytokines related to T cell migration and activation [52]. Recently, it has been shown that OCLs can induce immunosuppressive or inflammatory T cells depending on their origin. OCLs from healthy donors activate both CD4^+^ and CD8^+^ T cells, which produce the immunosuppressive cytokines IL-10 and TGF-β [53]. In a similar vein, OCLs differentiated from BM of healthy mice induce immunosuppressive FoxP3^+^ CD4^+^ T cells. Moreover, the same study proved that OCLs differentiated from mice with inflammatory bowel disease and bone loss activate tumour necrosis factor-α (TNFα)^+^ CD4^+^ T cells [54].

The diverse activities of OCLs may be directly involved in the development and progress of different diseases related to inflammatory and bone alterations. Moreover, a vicious cycle between inflammation and osteoclastogenesis may ensue since osteoclastogenesis is also promoted by inflammation-related cytokines that are associated with the pathogenesis of different arthritic diseases, such as rheumatoid arthritis or OA. Interleukin (IL) -1β is one of the major cytokines involved in OA and its effect on chondrocytes and synoviocytes has been widely studied [55,56], but its role in bone turnover remains unknown. IL-1β has been described as a regulator of OCL formation [57]. Recently, it has been shown that RANKL-expressing T reg cells were activated by IL-1β inducing OCL differentiation and bone loss [58]. However, it has also been demonstrated that IL-1β blocked the RANK/RANKL signalling in human OCP, preventing osteoclastogenesis [59]. As with IL-1β, TNF-α is also considered a key mediator of OA and it enhances OCL differentiation by multiple mechanisms. TNFα addition activates the RANK/RANKL pathway synergistically with other cytokines, such as RANKL or IL-1 [60,61], as well as increases RANK expression on OCPs [62] and RANKL production by osteoblast and stromal cells [63,64]. Independently of the RANKL signalling, TNFα can also directly induce OCP migration to the inflammation site [65] and OCP fusion [66,67], promoting OCL differentiation. The effect of TNFα on bone remodelling in inflammatory diseases has been proved by the employment of anti-TNFα biologics that have efficiently blocked osteoclastogenesis and reduced bone loss [68,69]. IL-6 and IL-17 also participate in the pathogenesis of the OA and their effect on bone turnover has recently been assessed. IL-6, together with TNFα, can increase osteoclastogenesis in a RANKL-independent manner [70,71]. IL-17 from patients with rheumatoid arthritis also induces OCL differentiation related to the increased RANKL production by mesenchymal cells [72]. Consistent with these results, Sato et al. [73] showed enhanced RANKL expression by mesenchymal cells and Th17 cells in the presence of IL-17. Therefore, increased osteoclastogenesis was observed in both in vitro and in vivo experiments [73], and blocking the activity of these cytokines ameliorated bone destruction [74,75].

Increased levels of RANKL and inflammatory cytokines in OA joints [76] and the presence of impaired OCPs could be linked to the enhanced osteoclastogenesis observed in OA patients, as it has recently been assessed. In comparison with healthy OCPs, peripheral blood mononuclear cells (PBMCs) from OA patients resulted in a high number of OCLs and greater bone resorption. These results were not related to an altered number of total CD14^+^ circulating cells, but they were explained by the reduced apoptosis of OCLs [77]. Employing a different gating strategy in the flow cytometry analysis of circulating OCPs, Loukov et al. found a lower number of circulating monocytes in OA women than in healthy controls. OA monocytes expressed elevated monocyte activation markers and produced higher levels of TNF and IL-1β. Moreover, C-C Motif Chemokine Receptor 2 (CCR2) was highly expressed in classical and intermediate monocytes [78]. This profile could boost the migration of OCP to the joint and promote OCL differentiation, since osteoblast and inflammatory cells secrete large amounts of C-C motif chemokine ligand 2 (CCL2) that promotes osteoclastogenesis and bone destruction [79]. Moreover, Shi et al. identified an increased expression of several genes related to the OCL differentiation pathway in PBMCs from OA patients, compared to healthy controls [80]. Altogether, these results provide evidence of the osteoclastogenic capacity of circulating OCPs in OA.

## 3. Current Treatments for Osteoarthritis

OA treatment continues to face important challenges in controlling symptoms and progression of the disease. Currently, treatment employed in OA patients cannot stop the degenerative process of this pathology and the available drugs are mostly focused on symptomatic therapy. Control of pain and amelioration of the joint function is mainly based on exercise programs and administration of paracetamol, nonsteroidal anti-inflammatory drugs, opioids, corticosteroids, and hyaluronan [81]. However, an individualized treatment is recommended since several results have indicated that administration of these treatments not only it is not well correlated to reduced OA symptoms, but it may also induce serious adverse events [82,83,84,85].

Other treatments based on symptomatic slow-acting drugs for osteoarthritis (SYSADOAs) are being considered for OA symptoms management. The most studied SYSADOAs are glucosamine (GS) and chondroitin (CS), and an excellent safety profile has been associated with these drugs [86]. Nonetheless, there is controversy over their effectiveness. A recent meta-analysis pointed out an improvement in the total Western Ontario and McMaster Universities Arthritis Index (WOMAC) by the combination of GS and CS. However, no effect was obtained in the visual analogue scale (VAS) [87]. Other authors have also claimed that GS and CS did not demonstrate superiority over the placebo in OA symptoms amelioration [88,89]. Due to these variable results, SYSADOAs may not be considered as first-line treatment for the management of OA.

Platelet-rich plasma (PRP) is defined as a volume of plasma with a platelet concentration in small volume of plasma higher than the average in peripheral blood [90]. Numerous studies have been conducted to analyse the employment of intra-articular injection of PRP in OA therapy. A meta-analysis of randomized controlled trials indicates that this treatment can reduce pain and improve the quality of life. However, the authors claimed that confirmatory trials are necessary to provide firm evidence [91]. Most recently, Dório et al. described no differences in pain and function in OA patients when PRP injection was compared with plasma and saline [92]. Similar results were found in the RESTORE randomized clinical trial where no differences between PRP and saline injections were observed in patients with symptomatic knee OA [93]. The lack of consistent results on PRP injection for the treatment of OA may be related to the different protocols and PRP products employed, and it makes it difficult to determine the clinical effectiveness of this therapy.

New treatments known as disease-modifying osteoarthritis drugs (DMODs) have been developed in order to slow OA progression, but the employment of these drugs has not yet been approved by either the US Food and Drug Administration (FDA) or the European Medicines Agency (EMA). DMODs includes varied mechanisms of action, such as targeting of pro-inflammatory cytokines or the Wnt pathway, among others [94,95]. As the number of potential DMODs is increasing, the intention of this work is not to review the different DMODs, but rather to overview the osteoclasts targeted therapy. Inhibitors of OCLs differentiation or activity have been clinically tested as DMODs in the treatment of OA but the effect of these molecules may also depend on the individual OA characteristics. Bisphosphonates bind to hydroxyapatite on bone of active remodelling areas. Therefore, these drugs are taken up by OCLs during bone resorption, interfering with various internal processes of these cells and, consequently, inhibiting bone loss [96]. Bisphosphonates have shown promising results for the treatment of OA in different animal models [97], but limited evidence confirms their clinical effects. While bisphosphonates may prevent OA progress when employed as early prevention treatment, they may not be effective in advanced OA [98,99]. Moreover, intravenous administration of bisphosphonates fails to reduce OA symptoms [100]. Improved bone quality by strontium ranelate administration is related to increased osteoblastogenesis and a reduction in bone resorption by altering the cytoskeleton organization of OCLs and inhibiting osteoclastogenesis [101,102]. Clinical studies with strontium ranelate have shown a structural improvement of OA patients, but its effect on the disease symptoms is still unclear [103]. Cathepsin K is a protease essential for bone resorption [104] and the inhibition of this enzyme has also shown favourable effects on the treatment of bone diseases [105,106]. However, the analysis of the cathepsin k inhibitor Odanacatib on postmenopausal osteoporosis ended due to the emergence of cardio-cerebrovascular adverse effects during the study [106,107]. A new cathepsin K inhibitor, MIV-711, has been tested in clinical trials as OA treatment. This drug improved bone and cartilage structure, analysed by magnetic resonance imaging (MRI), without inducing serious adverse events [108]. A second study confirmed the positive effect of this treatment on bone remodelling, and it also demonstrated pain relief for patients with unilateral knee pain [109]. Altogether, these data suggest that mitigation of bone destruction by reducing the OCL number or activity may improve patients’ conditions, therefore revealing the importance of undergoing further studies that address the effect of novel OA therapies on osteoclastogenesis.

## 4. Mesenchymal Stem Cell Therapy Targeting Osteoclast

Cellular therapies with MSCs are currently considered as essential in clinical research. MSCs are multipotent progenitors with the ability to transform other cell types in the right microenvironment. These cells are distributed throughout the body, allowing the continuous renewal and functional maintenance of all tissues [110]. In addition, MSCs have immunomodulatory capacity and very low alloreactivity, since they express very little MHC class I and they lack the MHC class II, CD86, CD40, and CD40L co-stimulatory molecules [111,112].

MSC-mediated immunomodulation acts on both innate and adaptative immune cells via both cell–cell contact and paracrine factors [113,114] and contributes to tissue regeneration. Monocytes are myeloid cells able to differentiate into either macrophages, dendritic cells, or OCLs [115]. As a part of the innate immune response, shared activities of these cells include phagocytosis and both inflammatory and tolerogenic responses, depending on the local environment. Maqbool et al. showed reduced monocyte differentiation into macrophages and dendritic cells by the presence of MSC. Moreover, either the phagocytic activity or the capability to induce T lymphocyte proliferation of each subgroup of myeloid cells were also blocked by MSCs [116]. On the other hand, the immunomodulatory effect of MSCs on monocytes and dendritic cells has also been reported in rheumatoid arthritis. Recently, it has been demonstrated that MSCs alter monocytes and dendritic cells towards an anti-inflammatory phenotype, and it may block the recruitment of monocytes and T cells into joints [117], reducing inflammation. In addition, the influence of MSCs on anti-inflammatory M2 macrophages has been widely analysed. It has been shown that different proteins released by MSCs, such as indoleamine 2,3-dioxygenase, prostaglandin E2 [118], TGF-β [118], and MSC-derived exosomes [119,120,121], induce an M1-to-M2 switch. Moreover, both MSCs and M2 macrophages have demonstrated to inhibit T cell proliferation and induce regulatory T cells [122]. Altogether, MSC-educated immune cells boost an immunosuppressive environment characterized by anti-inflammatory mediators such as IL-10 and TGF-β, while proinflammatory factors would be halted [113,114].

Besides the immunomodulatory capacity of MSCs, the local role that MSCs play in maintaining joint homeostasis, and their participation in the imbalance that occurs in a pathology such as OA, is now beginning to be understood. Dennis McGonagle et al. have described that joint-resident MSCs occupy several bone and joint cavity niches, including cartilage, synovium, infrapatellar adipose tissue, and synovial fluid. The advanced OA is associated with a numerical increase but functional decline in MSCs in determined regions, suggesting direct involvement of MSCs in OA [123]. For example, it is known that there is a small group of MSCs with chondrogenic capacity in normal superficial cartilage with CD166^+^ as a biomarker to identify [124,125,126]. In different articular pathologies, such as OA, these MSCs enhanced the migration and proliferation [127,128]. It is known that during bone turnover OCLs participate in the process of migration and differentiation of bone marrow-derived mesenchymal stem cells (BMSCs), enhancing their recruitment and differentiation into osteoblasts [28,29].

During the last decade MSCs have become very promising as novel therapeutic agents for a wide range of pathologies [129], and in particular for degenerative conditions in which there is an essential inflammatory component as well as defects in bone remodelling, such as osteoporosis, rheumatoid arthritis, and OA. However, the use of MSCs as a clinical therapeutic tool for the treatment of diseases is very complex. Firstly, there are many sources of MSCs (BMSCs, adipose tissue (AD-MSCs), umbilical cord, dental pulp, placenta, amniotic fluid, etc.) [130]. Secondly, in pre-clinical and clinical assays, MSCs are administered after isolation and expansion in cell culture or as stromal vascular fraction (SVF) [131,132]. The SVF niche contains different subsets of MSCs, depending on the type and size of the blood vessel and tissue (MSCs precursors of endothelial cells, fibroblasts, etc., and small fraction of AD-MSCs) [133]. Thirdly, depending on where the MSCs are obtained, they have different therapeutic potential for one or another clinical problem (proliferative, immunomodulatory, profile of cytokines and growth factors release, etc.) [134]. Indeed, many factors, such as the storage temperature of the MSCs and preservation agents employed, could also affect the effectiveness of this cell therapy [135,136]. For these reasons, we are still far from knowing the most optimal conditions for the use of MSCs as a therapeutic tool in preclinical and clinical trials in OA [137,138]. Despite numerous existing studies, no significant clinical effects of these cells on OA have been determined, although some authors describe an improvement in the observation of the clinical image and histological results after the infiltration of MSCs in the knee [11,139].

Currently, the most widely used exogenous source of MSC as autologous clinical therapy is AD-MSC, since it is an abundant tissue in the body, easily accessible, with a greater proliferative and immunomodulatory capacity than those derived from bone marrow. AD-MSCs are an excellent source of cells for the treatment of OA [140,141]. However, the mechanism by which AD-MSCs induce beneficial effects in joint is not yet determined [142]. AD-MSCs are known to regulate the local microenvironment mainly via paracrine factors rather than cell–cell contact. The secretion of bioactive molecules is a more effective mechanism for local regulation since their easy distribution makes the effect more extensive [143,144].

Evidence regarding the involvement of MSC in OA are mainly based on in vitro studies using chondrocytes, synoviocytes, or osteoblasts [145,146,147,148]. However, there are several recently published articles that have been aimed at describing the effect of MSCs on OCL differentiation. Furthermore, various studies have attempted to unravel the mechanisms underlying these actions of MSCs. The results obtained concerning the impact of MSCs on osteoclastogenesis have been controversial.

Several pieces of in vitro work using MSCs of different origins, such as AD-MSCs or BMSCs, have shown that these cells may act by inhibiting the differentiation of human OCPs into mature OCLs. This effect has been assessed by direct counts of the osteoclast-like TRAP^+^ cells in co-cultures, supported by expression analysis of OCL-specific markers such as cathepsin K and NFATc1, as well as functional tests such as bone degradation assays [129,149,150,151]. The requirement of cell-to-cell contact for this blockade of osteoclastogenesis in vitro has been discussed. In this sense, using diverse strategies, such as transwell experiments [151] or conditioned medium obtained from MSCs [98], several authors concluded that direct contact is not necessary for the MSC-mediated inhibition of osteoclastogenesis. However, other reports [150] claim the existence of two different pathways that may be participating in the negative role of MSCs on OCL formation, one being contact dependent and the other being mediated by soluble factor secretion.

Among the mechanisms that have been explored, a feasible participant for the inhibitory effect of MSCs on OCL differentiation is osteoprotegerin (OPG). This decoy receptor for RANKL blocks the RANKL–RANK interaction and therefore inhibits osteoclastogenesis. In vitro assays have demonstrated that MSCs produce and secrete OPG. Moreover, both the treatment with anti-OPG neutralizing monoclonal antibodies in co-cultures and the transfection of human MSCs with OPG siRNA, partially recover the OCL differentiation. This evidence points out that the MSC-mediated blockade of osteoclastogenesis is in part OPG dependent [129,149].

Other mediators that may indeed participate in the inhibitory effect that is being discussed are CD39 [151] and CD200 [150]. In the case of CD39, which is an ectoenzyme that hydrolyses ATP, its implication in the ADSCs-mediated inhibition of osteoclastogenesis was determined using POM1, an inhibitor of CD39, which almost completely abolished the observed effect [151]. Regarding the role of CD200 in the blockade of osteoclastogenesis, in vitro studies were carried out using recombinant CD200 in the cell cultures. Moreover, the implication of CD200 was also confirmed by carrying out co-cultures using two previously isolated fractions of MSCs: CD200^+^ and CD200^−^. Both fractions expressed similar levels of OPG (therefore excluding the implication of this protein in this experimental setting), but the reduction in OCL activity was only observed in co-cultures between MSC CD200^+^ and PBMC adherent cells and not in the case of co-cultures using CD200^−^ cells.

The immunosuppressive effect of MSCs could also affect osteoclastogenesis since MSCs act on different immune cells and can regulate OCPs [122]. OA joints are characterized by an inflammatory environment [55,56] that can induce OCL differentiation and the tolerogenic activity of MSCs may change the profile of lymphocytes and leukocytes reducing joint damage. In vitro, MSCs act on monocytes and dendritic cells, inhibiting the production of inflammatory cytokines and reducing chemotactic factors [117,152,153].

Nevertheless, studies showing the opposite effect of MSCs on OCL differentiation can also be found in the literature. In this regard, promotion of osteoclastogenesis has been shown in vitro using several experimental settings [154,155,156]. Mbalaviele et al. [154] used a co-culture framework to study osteoclastogenesis in the absence of exogenous cytokines, growth factors, or hormones. Their results show that TRAP^+^ cells were obtained from human hematopoietic stem cells (HSCs) when co-cultured with human bone-marrow-derived MSCs. This effect was not observed in the absence of human MSCs, nor when co-cultures were carried out using HSCs and other cell types such as fibroblasts. Moreover, cell contact was shown to be determinant in order to observe the mentioned effect. Interestingly, the authors state that, in this context, HSCs may regulate the MSCs production of cytokines involved in osteoclastogenesis.

These contradictory in vitro results regarding the actions of MSCs on osteoclastogenesis may derive from the diversity in the utilized experimental approaches. The absence of exogenous growth factors and cytokines [154], or the use of diverse cell types in the co-cultures employed to analyse OCL differentiation (such as differentiated osteoblasts [156,157], or rat OCPs [155], may determine this variability in the conclusions drawn by the mentioned studies.

### 4.1. Clinical Evaluation of MSCs Therapy

The aforementioned in vitro experiments point out the feasible role of MSCs on OCL differentiation, which is one of the events that participates in the development of OA. Regarding the use of these cells in clinical settings aimed at ameliorating the disease progression in OA patients, there are several trials that have been published recently. The safety of this cell therapy and the assessment of adverse effects have been analysed, with a number of them showing a good overall safety profile [158,159,160]. These findings have indeed sustained a variety of other clinical trials that focus on the efficacy of this therapeutic use of MSCs, looking mainly at the pain and functional changes evaluated through a variety of tools such as the WOMAC score, VAS, and knee injury and osteoarthritis outcome score (KOOS index), among others. Furthermore, structural assessment has also been included in these clinical trials by performance of MRI and histological analysis.

The features and key characteristics of the different clinical trials are variable (Table 1). However, the results show an overall promising effect of the treatment with MSCs in OA patients, reducing the clinical and pain scores significantly [160,161,162,163,164]. The full implementation of the therapy with MSCs for OA patients in clinics needs further analysis and standardization of protocols.

In this sense, one of the factors that has been discussed is the source of MSCs. As Jo et al. [159] claimed in their study published in 2014, AD-MSCs show several technical advantages in comparison to other sources, such as BMSCs. The feasibility of harvesting in a large amount using a minimally invasive method, the rapid expansion in culture, and the lower effect of age or morbidity of patients on cell quality are some of the benefits of this source of MSCs [159]. Moreover, the loss of homing ability of the different types of MSCs has also been considered [129], as well as the potentially stronger immunomodulatory properties of AD-MSCs compared to cells from other tissue sources [160].

The dose, number of injections, and route of delivery are other factors that vary among the different clinical trials. These are essential points that need further clarification to ensure efficacy of the therapy while avoiding possible negative effects such as fibrous foreign body formation [159,160] or other technical and clinical complications. Moreover, the follow-up period is also variable among the published clinical studies and there is no uniformity regarding the implementation of a period of non-weight bearing after injection or a standardized rehabilitation protocol. The primary and secondary outcomes regarding the efficacy and safety of MSC injection in OA patients have been analysed in comparison to control placebo groups [164], but also using active controls, such as hyaluronic acid [162], or in combination with co-adjuvants, such as platelet rich plasma (PRP) [163]. Regarding the use of combined treatment protocols, further analysis shall be carried out in order to determine their safety and efficacy profiles. Indeed, several of the clinical trials that have been undertaken using MSC for OA treatment imply the discontinuation of pain medication (except rescue analgesics) in the participants [159,164]. Furthermore, certain clinical studies [160] apply exclusion criteria consisting in the previous administration of oral/intra-articular corticosteroids and injection of hyaluronic acid derivatives that are recommended therapies for patients with knee OA [81,167]. However, other cases, in which studies have been performed maintaining the analgesics used as part of their routine management [161], or actively integrating MSC therapy with other treatments such as hyaluronic acid or PRP injections [162,163], may be useful to determine the feasibility of such combinations.

Another limiting factor that will need to be considered for the safety evaluation of this cell treatment is the clinical condition of the patient. Recently, the potential association between MSC therapy-related adverse events and certain risk factors, such as age, gender, or pathologies, has been pointed out [168]. Thus, MSC administration should be performed considering the main risk factors of the patient to assure a good safety profile.

The mechanisms of action underlying the observed effects of MSCs injection in OA patients are still under discussion. The amelioration of the disease progression in the participants of these clinical trials are probably due to several biological effects. In the first place, MSCs can act by exerting paracrine effects that lead to a modulation of the inflammatory environment and could also be acting through the promotion of a chondro-protective milieu [160,163]. Therefore, MSCs may be contributing to the observed functional improvement and pain relief, both directly by releasing bioactive mediators and indirectly by affecting the cytokine and growth factor production from endogenous cells [159]. Another possible mechanism of action—as pointed out in the literature—is based on the potential of MSCs to differentiate into chondrocytes and therefore promote cartilage regeneration in these patients. There is still controversy regarding this regenerative effect that is indeed observed in some clinical settings [159,162] while absent in other studies [163,164]. These differences may be due to the participation of patients with a higher degree of OA and therefore a scarce number of viable chondrocytes, or other factors such as the duration of the follow-up period. Nevertheless, other authors claim that the effect of MSCs is probably not due to their direct differentiation into regenerative or replacement tissue [169]. Lastly, MSCs may alter the PBMC composition and reduce its inflammatory profile in OA patients, which could also affect bone marrow cellularity and, consequently, bone remodelling. The intra-articular injection of AD-MSCs in OA patients alters the proportion of circulating OCPs, which could reduce OCL differentiation and bone turnover in this pathology [165]. Moreover, BMSCs treatment has also been shown to reduce CD14^+^CD16^+^ OCPs in the synovium [166].

Therefore, an additional possibility that needs to be addressed is the potential effects of MSCs on OCL differentiation and/or recruitment of OCPs to the joint in this clinical context. This ulterior mechanism of action is not considered in the published clinical trials, though feasible. The participation of the MSC-mediated actions on OCL differentiation and their contribution to the observed beneficial effects of MSCs therapy in OA patients is yet to be dealt with [170].

### 4.2. MSC-Derived Extracellular Vesicle Therapy

The number of publications that refer to extracellular vesicles (EVs) as a possible new therapeutic tool or as early and evolutionary markers of pathology has grown exponentially. EVs are signalling vesicles released by cells, including MSCs. Currently, studies indicate that perhaps the most effective way for cells to communicate is not through a cell–cell contact mechanism but through their secretome that contains different molecules and EVs. In 2018, the International Society for Extracellular Vesicles (ISEV) defined EVs as the generic term for particles naturally released from the cell that are bounded by a lipid bilayer and cannot divide because they have no nucleus [171]. EVs modulate the activity of other cells through the molecules that they carry. It is interesting to note that EVs from MSC have very low immunological capacity [172] although they can transport major histocompatibility complexes and activate T cells [173]. It is essential to consider EVs derived from MSCs as potential new therapeutic agents without the use of cells [174].

Two types of EVs have been described based on their biogenesis: “exosomes or small EVs” of endosomal origin, and “microvesicles or medium/large EVs” derived from the cellular membrane. Due to their complex biogenesis, it is very difficult to determine specific markers [171]. The most used differentiating criterion is the particle size: small EVs (<200 nm) and medium/large EVs (>200 nm). The membranes of the medium/large EVs come from the plasma membrane of the cell where they originate, so they could have proteins or structures of extracellular origin attached to them. These exogenous molecules could influence the effects that the medium/large EVs may exert. The membranes of the small EVs do not present this problem as they have an intracellular origin [175]. There are also many differences between the procedures used to isolate and quantify EVs, in the techniques for studying vesicular content (proteins, lipids, nucleic acids, etc.), lipids that can be co-isolated together with EVs, forms of storage, specific or common functions, etc. Unfortunately, all of this means that the published in vitro and preclinical results are not comparable since there is a great methodological variability. The standardization of isolation and characterization methods is an essential step for the development of research into EVs as therapeutic agents or pathology markers. Currently, small EVs have become the main type of EVs studied as therapeutic agents due to their intracellular biogenesis [175].

EVs are a potentially better alternative to MSC therapy without the clinical risks associated with cell therapy, with a more precise dosing and treatment control, being less microenvironment-dependent, and with more predictable effects. In addition, the small size is an advantage in relation to the selection of routes of administration and the ease of internalization in the target cell, since they are vesicles formed by lipid membranes derived from other cells [176]. The beneficial effects of EVs on OA have been demonstrated in vitro. In primary cultures of IL-1β-stimulated OA osteoblasts both small and large EVs of AD-MSCs decrease senescence markers, inflammation, and oxidative stress, and normalize the mitochondrial membrane potential of these cells [177]. A chondroprotective role of AD-MSC-derived EVs on IL-1β-stimulated OA chondrocytes in primary culture also has been demonstrated [178]. IL-1β reduces the expression of the antioxidant enzyme Prdx6 in OA chondrocytes compared to unstimulated ones, and this effect is reversed by medium/large EVs. Interestingly, it was demonstrated that Prdx6 is found in high concentration in these AD-MSCs-derived EVs. This suggests the chondroprotective potential of the medium/large EVs [179].

The same occurs in preclinical trials with experimental models in vivo, with naïve or modified EVs, administered by local infiltration in different animal species [180]. EVs derived from human MSCs of different cellular origin produce effects of neovascularization, and bone and cartilage regeneration in bone and joint pathologies. In a mouse model of OA, this treatment protected articular cartilage and gait abnormality by increasing the expression of collagen II [181]. Implicated mechanisms include the transfer of molecules such as miRNAs, mRNAs, and proteins to target cells. However, it is still too early to be able to claim that these tests have a significant effectiveness due to the methodological differences previously exposed with respect to the isolation, characterization, and cellular origin of the EVs that have been infiltrated, or even if the positive effect achieved is maintained long term.

Curiously, the number of reviews regarding the beneficial potential of MSCs and small EVs in OA pathology has increased very recently [137,141,182,183]. These reviews indirectly suggest that there may be a modulatory action of MSCs or of MSC-derived EVs on OCL differentiation in osteoarticular pathologies. However, there are only a handful of them in which the direct effect of EVs on OCLs has been studied. In a murine model of OA of the lumbar facet joint, Jinsong Li et al. [184] demonstrated that small BMSCs EVs reduce nerve infiltration and angiogenesis in subchondral bone, significantly blocking the increase in TRAP^+^ cell population and OCL activity. Small EVs inhibit osteoclastogenesis by suppressing the RANKL–RANK–TRAF6 signalling pathway in subchondral bone [184]. On the other hand, it has recently reviewed that EVs are able to inhibit inflammatory cell migration and inflammatory cytokine production in different in vitro and in vivo models of inflammatory and degenerative diseases [113]. This immunomodulatory activity of EVs could ameliorate bone diseases by decreasing OCP recruitment and fusion. Unfortunately, there are not enough published works that refer to the effects of EVs from MSCs on OCLs that can help us achieve a better understanding of the relationship between these two cell types in OA.

The content of EV is also the object of therapeutic study. Non-coding RNA (ncRNA), such as miRNA, IncRNA, circRNA, and siRNA, are molecules that regulate the transcription and translation of proteins and are transported by EVs. A recent review mooted the idea that the ncRNAs of small EVs secreted by osteoblasts, OCLs, chondrocytes, and other cells inhibit OCL differentiation, enhance chondrocyte activity, and promote angiogenesis in osteoporosis, bone fracture, and OA [185]. It is interesting to note that, so far, of the osteoarticular pathologies studied, only in osteoporosis it has been seen that the ncRNAs contained in the EVs produced by the cells present in the joint had effects on OCLs. Thus, Song et al. [186] determined that miRNA-155 derived from small EVs of endothelial cells can inhibit the activity of OCLs. miRNA-155 activity is induced on OCPs by interferon-β [187] and TGF-β1 [188], and its negative effect on OCL differentiation may be related to the inhibition of suppressor of cytokine signalling (SOCS1) and microphthalmia-associated transcription factor (MITF)—two key regulators of osteoclastogenesis. At this time, due to the very few studies linking the effects of ncRNAs on OCLs, it is difficult to make a prediction as to how the role of OCLs via this mechanism might affect OA therapy.

## 5. Concluding Remarks

Joint cells are in permanent direct contact with local MSCs in the joint, with regulatory signals being transferred in both directions. Alteration of any of these cells can lead to conditions such as OA. Great efforts are being made to design new MSC treatments with the aim of restoring OA joint function. Several clinical trials have demonstrated the potential efficacy of MSCs derived from bone marrow, adipose tissue, and umbilical cord blood in the treatment of OA. However, the mechanisms by which MSCs improve the pathogenesis of OA remain unknown. Most research is focused on chondrocytes and cartilage improvements, but there is very little literature that refers to the therapeutic effects of MSCs on OCLs. Osteoclastogenesis is increased in OA and subchondral bone loss is enhanced in this pathology. OCL activity induces MSC migration and osteoblastogenesis. This effect may improve bone quality, but it could also facilitate the recruitment of MSCs into the joint. Additionally, MSCs may affect OCL differentiation through cell–cell contact or by the production of soluble factors and extracellular vesicles. These mediators may alter OCP activation and their migration into the bone marrow, as well as they could reduce inflammation and block OCP fusion and, therefore, OCL maturation. Thus, improvement of OA by MSC therapy may be not only related to its chondroprotective effect but also to its blocking activity on OCLs. However, the opposite results obtained by other studies demonstrating an osteoclastogenic activity of MSCs suggest that more complex interactions may exist between these two articular cells (Figure 1). Future work addressing this issue is expected to clarify the particular mechanism through which MSCs or MSC-derived EVs may be affecting OCL activity in OA pathology, therefore providing alternative strategies for the development of novel and more effective therapies for OA.

## Figures and Tables

**Figure 1 ijms-23-04693-f001:**
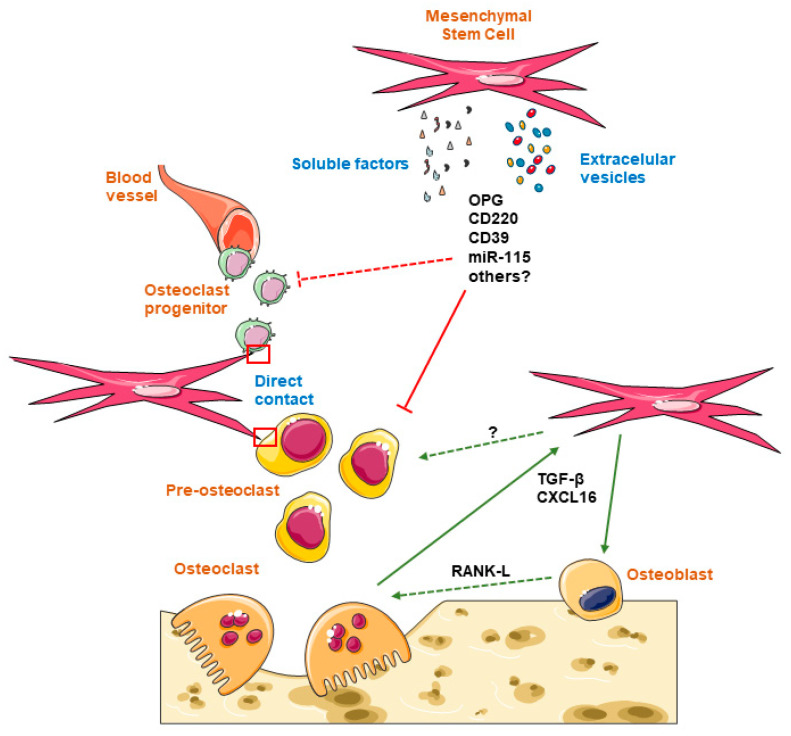
Possible interactions between mesenchymal stem cells and osteoclasts. MSCs could block osteoclastogenesis through either direct contact or production of secretome, including soluble factors and extracellular vesicles. Mediators, such as OPG, CD220, CD39, and miR-115, may be implicated, inhibiting the osteoclast progenitors’ recruitment (˫ – –) and/or pre-osteoclast fusion (**˫**⸺). However, the opposite effect on osteoclastogenesis cannot be excluded (‹– –). Another mechanism to be considered is the role of osteoclasts in mesenchymal stem cell migration and differentiation into osteoblasts through the release of TGF-β and/or CXCL16 (**→**). Altogether, these interactions should be taken into account in the analysis of the novel approaches based on MSC therapy in osteoarthritis.

**Table 1 ijms-23-04693-t001:** General characteristics of clinical trials using MSCs for OA treatment.

Study	Intervention	Primary Endpoint	Secondary Endpoints	Comparison Group
Jo CH et al., 2014 [159]	Intra-articular injection of AD-MSC	Safety and function WOMAC index	Clinical, radiological, arthroscopic and histological evaluations	Baseline, dose groups
Pers YM et al., 2016 [160]	Intra-articular injection of AD-MSC	Safety of dose-escalation	Clinical efficacy (pain and function scales)	Baseline, dose groups
Al-Najar M et al., 2017 [158]	Intra-articular injection of BMSCs	Safety	Clinical efficacy (function score and structural component)	Baseline
Pers YM et al., 2018 [165]	Intra-articular injection of AD-MSC	Profile of immune cells in fresh peripheral blood	Pain and function index	Baseline
Freitag J et al., 2019 [161]	Intra-articular injection of AD-MSC	Safety, pain and functional changes	Disease modification detected through radiological examination	Conventional conservative management
Lu L et al., 2019 [162]	Intra-articular injection of AD-MSC (Re-Join^®^)	Function WOMAC index	VAS scale, radiological outcomes and safety profiles	Hyaluronic acid injection
Lee WS et al., 2019 [164]	Intra-articular injection of AD-MSC	Function WOMAC index	Clinical and radiologic examination, and safety.	Saline injection
Chahal J et al., 2019 [166]	Intra-articular injection of BMSCs	Safety	Clinical, radiological, and biomarkerassessments	Baseline, dose groups
Lamo-Espinosa JM et al., 2020 [163]	Intra-articular injection of BMSCs + PRPGF	Pain and function scores	Radiological assessment	Platelet Rich Plasma (PRGF^®^)

## Data Availability

Not applicable.

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
