# Peer review of "Connection between Mesenchymal Stem Cells Therapy and Osteoclasts in Osteoarthritis"

_ijms, 2022, doi:10.3390/ijms23094693_

Round 1

Reviewer 1 Report

IJMS-1689328

In this work, the author reviewed the fundamentals and the state-of-art of MSC therapy of osteoarthritis. The article is informative and well written. From the perspective of academic criticism, several technical concerns need to be addressed to further improve the quality of this manuscript, as appended below.

The review was not very well structured, and the title/subtitles were ambiguous:

  • In the “1. Crosstalk between articular cells in osteoarthritis”, the author introduced the fundamental of OA pathogenesis and the cellular crosstalk involved. It would be better to structure the section into 2 sub-sections: 1. Fundamentals of osteoarthritis; 2. Cellular crosstalk in OA. It would also be nice to extend the content to a 3rd sub-section focusing on the role of MSCs in the osteoblast development and the microenvironment to provide more context for the later discussion of MSC therapy.
  • The section “3. Osteoclasts as potential therapeutic target in osteoarthritis” should be renamed by a title more related to MSC therapy, like “MSC therapy targeting OCLs”. The sub-section, “3.1. Antiresorptive agents” should be changed to a sub-section discussing the existing therapeutic methos that are not involving MSCs or any stem cell products, and address the pros and cons. Otherwise, the author should consider changing the title and main focus of the manuscript.
  • The discussion of EVs should be an extension of MSC therapy as the next-generation MSC therapy. So changing the sub-section title to something like “MSC-derived EVs therapy” or “Next-gen MSC therapy” would be more appropriate.

As briefly mentioned in the review, the interaction between monocyte/macrophage and MSC contributes significantly to the OA regeneration, and the immunomodulation of MSCs is also an important aspect in their therapeutic effects. It would help to complete the story if a section extending this discussion is included in the review.

Author Response

1. Comment: English language and style are fine/minor spell check required. Answer: The grammatical errors have been checked and corrected.

2. Comment: In the “1. Crosstalk between articular cells in osteoarthritis”, the author introduced the fundamental of OA pathogenesis and the cellular crosstalk involved. It would be better to structure the section into 2 sub-sections: 1. Fundamentals of osteoarthritis; 2. Cellular crosstalk in OA. It would also be nice to extend the content to a 3rd sub-section focusing on the role of MSCs in the osteoblast development and the microenvironment to provide more context for the later discussion of MSC therapy.

Answer: The section 1 has been divided in two sub-sections, 1.1. Fundamentals of osteoarthritis (line 30) and 1.2. Cellular crosstalk in osteoarthritis (line 57), as reviewer 1 has proposed. Moreover, the role of MSCs in the osteoblast development and the microenvironment has been discussed (lines 83-97).

3. Comment: The section “3. Osteoclasts as potential therapeutic target in osteoarthritis” should be renamed by a title more related to MSC therapy, like “MSC therapy targeting OCLs”. The sub-section, “3.1. Antiresorptive agents” should be changed to a sub-section discussing the existing therapeutic methos that are not involving MSCs or any stem cell products, and address the pros and cons. Otherwise, the author should consider changing the title and main focus of the manuscript.

Answer: The section “3. Osteoclasts as potential therapeutic target in osteoarthritis” has been divided in two sections: “3. Current treatments for osteoarthritis” (line 215) and “4. Mesenchymal stem cell therapy targeting osteoclast” (line 281). The section “3. Current treatments for osteoarthritis” discuss the existing therapeutic methods that are not involving MSCs or any stem cell product, as requested by reviewer (lines 223-254).

4. Comment: The discussion of EVs should be an extension of MSC therapy as the next-generation MSC therapy. So, changing the sub-section title to something like “MSC-derived EVs therapy” or “Next-gen MSC therapy” would be more appropriate.

Answer: The sub-section “3.2. Mesenchymal stem cells therapy” has been changed to section “4. Mesenchymal stem cell therapy targeting osteoclast” (line 281). Moreover, the sub-sections “3.2.1. Clinical evaluation of MSCs therapy” and “3.3. Extracellular vesicle therapy” have been included as sub-sections “4.1. Clinical evaluation of MSCs therapy” (line 411) and “4.2. MSC-derived extracellular vesicle therapy” (line 493).

5. Comment: As briefly mentioned in the review, the interaction between monocyte/macrophage and MSC contributes significantly to the OA regeneration, and the immunomodulation of MSCs is also an important aspect in their therapeutic effects. It would help to complete the story if a section extending this discussion is included in the review.

Answer: A section discussing the immunomodulatory role of MSC have been included (lines 289-309).

Reviewer 2 Report

The authors presented an interesting study concerning the connection between mesenchymal stem cells therapy and osteoclasts in osteoarthritis. The review is well presented and prepared; however, some remarks should be addressed.  

  • What are the risk factors of osteoarthritis? Does any factor eliminate or limit the application of mesenchymal stem cells therapy and osteoclasts?
  • Is mesenchymal stem cells therapy and osteoclasts dependent on the temperature? 
  • the activity and usability of mesenchymal stem cells therapy and osteoclasts in osteoarthritis should be presented in the form of a table, also include specific methods and markers that would enable the evaluation of the therapeutic efficacy efficacy
  • What do you think about combined protocols with steroid or hyaluronic acid? 

Author Response

1. Comment: What are the risk factors of osteoarthritis? Does any factor eliminate or limit the application of mesenchymal stem cells therapy and osteoclasts?

Answer: Information regarding the risk factors of osteoarthritis has been added (lines 31-47). Moreover, their implication in MSC therapy has also been addressed (lines 460-464).

2. Comment: Is mesenchymal stem cells therapy and osteoclasts dependent on the temperature?

Answer: Thank you for this comment. It is known that storage temperature is a key factor for the effectiveness of MSC therapy. Thus, this observation has been included in lines 334-336).

Comment: The activity and usability of mesenchymal stem cells therapy and osteoclasts in osteoarthritis should be presented in the form of a table, also include specific methods and markers that would enable the evaluation of the therapeutic efficacy.

Answer: A table showing the main characteristics of the clinical studies that we have cited in the review, which use MSC therapy for OA treatment has been included. This points out the similarities and differences among them and the different approaches in order to assess the activity/usability of MSC in these patients. (Table 1 in line 423 and 429).

3. Comment: What do you think about combined protocols with steroid or hyaluronic acid?

Answer: A paragraph addressing this issue has been included in lines 448-459.